# Extracellular Vesicles and Their Membranes: Exosomes vs. Virus-Related Particles

**DOI:** 10.3390/membranes13040397

**Published:** 2023-03-31

**Authors:** Daniela Cortes-Galvez, John A. Dangerfield, Christoph Metzner

**Affiliations:** 1AG Histology and Embryology, Institute of Morphology, University of Veterinary Medicine Vienna, 1210 Vienna, Austria; 2Celligenics Pte Ltd., 30 Biopolis Sreet, Singapore 138671, Singapore; 3Molecular Biotchnology, FH Campus Vienna, 1100 Vienna, Austria

**Keywords:** exosome, extracellular vesicle, virus, envelope, vesicle, membranes

## Abstract

Cells produce nanosized lipid membrane-enclosed vesicles which play important roles in intercellular communication. Interestingly, a certain type of extracellular vesicle, termed exosomes, share physical, chemical, and biological properties with enveloped virus particles. To date, most similarities have been discovered with lentiviral particles, however, other virus species also frequently interact with exosomes. In this review, we will take a closer look at the similarities and differences between exosomes and enveloped viral particles, with a focus on events taking place at the vesicle or virus membrane. Since these structures present an area with an opportunity for interaction with target cells, this is relevant for basic biology as well as any potential research or medical applications.

## 1. Introduction

Both cells of eukaryotic and prokaryotic origin produce extracellular vesicles (EVEs, for an overview of vesicle types and abbreviations, see Table 1). In bacteria, they contribute to horizontal gene transfer, waste disposal, organization of biofilms, and general inter-cell communication [1,2]. EVEs constitute the vesicular fraction of the cellular proteome [3]. Subfractions include microvesicles (MIVs), apoptotic bodies (ABOs), also collectively referred to as ectosomes for their cell wall provenance, and exosomes (EXOs, derived from intercellular membranes) [4]. Previously, nomenclature has often caused confusion in the field but in more recent times there has been some consolidation [5,6,7,8]. Several reviews are available, assessing the relationship of the different types of EVEs with different virus species [1,9,10,11,12,13]. In cells infected with an enveloped virus (EVI, characterized by a lipid bilayer membrane outer shell—the envelope), such as human immunodeficiency virus 1 (HIV-1), Influenza A virus (IAV), herpes simplex virus (HSV) or severe acute respiratory syndrome-associated coronavirus 2 (SARS-CoV-2), the secreted viral particles may also be considered extracellular vesicles albeit in a pathological context (see Figure 1 and Table 2). Both exosomes and membrane lipids have a strong influence on the viral lifecycle and vice versa [1,14,15]. Virus particles may be of ectosomal (HIV-1) or exosomal (HSV) origin [4]. In terms of size and core function (transfer and protection of cargo), EVIs are most similar to EXOs (see also Figure 1) [1]. The biggest difference is that while EXOs are an integral part of the physiological context of (multi)cellular organisms, EVIs are considered obligatory parasites outside of the same context. 

In eukaryotic cells, signal transfer is the predominant activity of EVEs, especially exosomes (EXOs). EXOs are among the essential sub-types of EVEs (together with MIVs and ABOs). EXOs have average diameters between 30 and 300 nm and contain cell-derived proteins and different RNA species [16], as can also be seen in Figure 1). The proteins may be soluble and found in the lumen of the vesicle or membrane associated by different tethering structures. The complete set factors (protein and RNA) is often termed “cargo” and refers to the fact that such elements may also be artificially incorporated into EXOs and constitute a form of payload, which at least partially explains the popularity of EVEs in biotechnology and biomedicine applications [16]. Functionality is therefore determined by the cargo vesicles. EXOs are a highly modular and thus flexible system (providing platform and scaffold) for cell-to-cell communication (see Figure 2). As a result, dynamic behavior is a consequence of exosome function [17]. The central issue for a virus is maintaining its DNA or RNA genome and avoiding any loss of information. A certain degree of permanence is required to achieve this. Viral proteins need to be part of the “vesicle”, both for structural and enzymatic functions in defined amounts and ratios. Additionally, the continuity of virus-replicative events is essential since a failure to produce new virions will essentially disrupt the viral lifecycle. To circumvent such potential pitfalls and the stringent imperative of never-ending replication, the virus may enter a state of latency. Virus species have adopted different methods to protect their genetic material from damage. In all viruses, a proteinaceous structure called the capsid surrounds the viral DNA or RNA. In a subset of virus families, a second layer of protection is provided by a cell-derived lipid bilayer—the envelope (viral envelope, VEN)—encompassing the capsid. In between the capsid and envelope, protein-rich matrices may be found (see Figure 1). In addition to protective duties, the envelopes also contain proteins of viral origin, responsible for attaching and invading target cells (often termed envelope or spike proteins). EVI particles share a similar size range to EXO and enter cells with the intention of re-programming cellular pathways to accommodate their own replication. While this re-programming is usually on a larger scale, it resembles the triggering of an intracellular signal transduction by EXO. While EXOs entering a recipient cell do not automatically trigger the release of a new set of EXOs, infection with an enveloped virus is predominantly aimed at replication and producing progeny. Ultimately, both lead to an altered state of expression in the target cells, i.e., a proteomic profile. Similarities become more obvious when considering the EXO and EVI lifecycle (Figure 2). So, is an EVI just an EXO with a capsid, acquiring viral proteins and nucleic acid and going to “the dark side”? The “trojan exosome hypothesis” suggests a potentially similar situation for HIV [10,12]. Morphologically, the presence of the highly structured capsid or core is the most prominent difference; however, an enveloped virus may be infectious without a core or capsid [18]. Significant overlaps in biochemistry and molecular biology are obvious, however, the complexity of viral envelope generation (see Figure 3) suggests a rather more diverse provenance. While the cellular contributions to the EXO/EVI lifecycle resemble each other, as well as the functions ascribed to EXO/EVI membranes, initiation and subsequent aspects differ (see Figure 2). The cellular producer needs to provide the biogenesis machinery (assisted by viral proteins in the case of EVIs) and building materials, including incorporated signals. The assembly of the EVE is cell-dependent regarding packaging and release. The transfer of the signal package to the recipient requires protection from physical and biochemical interference, which is provided by the vesicle membrane. The recipient cell needs to decipher the signals and consequentially start re-programming. The function of the EVEs membrane in this step is to provide attachment to cells and target specific subsets by defining the permissivity of cells for the EVE. The starting impulse (signaling for EXOs, virus propagation for EVIs) is different as is the outcome: virus particles need a certain degree of stability to maintain genome integrity, whereas EXOs need to reflect the signaling need, thus generating higher diversity (see Figure 2). 

Several groups of non-enveloped viruses use EVIs to escape immune surveillance or to broaden their infection range [19]. For virus particles, an evolutionary imperative exists, demanding the continuous replication and maintenance of the complete and intact genetic material—self-perpetuating and with a high degree of reproducibility. Such a strict mindset is not useful for EXO, since the adaptation to new environmental factors may need to change composition and cargo quicker. 

## 2. The Biogenesis of EXO and EVI

The formation of both EVIs and EXOs is characterized by the process of vesiculation, facilitated by cell-derived membranes. The bio-mechanical hallmark of these processes is a highly notable increase in membrane curvature, enabling the submicron diameter of the vesicles. Fundamental biophysical principles apply to the generation of liposomes and exosomes, as well as viral envelopes [20]. Essentially, vesicle formation consists of two steps: membrane deformation (wrapping a membrane around the future content) and membrane scission (pinching off the vesicle from a larger body of membrane). A degree of variation as to the provenance of viral envelope membranes is reported [21,22]. Viral capsids are usually preformed (by self-assembly or scaffold-mediated processes), however, these may not be fully developed since a step of viral maturation may be completed only after the exit of the virion from the cell. Mechanistically, envelopes are either generated by concomitant vesiculation and direct release at the cell membrane (ectosomal), a process termed budding, or by interaction with exocytosis-pathways at internal membrane structures (exosomal). Additionally, processes leading to cell lysis may take place which also facilitates viral release. Capsids may be formed in different compartments from nuclei to cytosol and this formation usually occurs in connection with internal membranes which provide enhanced mechanical strength. While *Retroviridae* primarily assemble at the plasma membrane, herpesviruses initially assemble in the nucleus, and transiently acquire a VEN at the nuclear membrane before acquiring their final envelope along the Golgi apparatus prior to release at the plasma membrane. This heterogeneity makes them somewhat difficult to compare (see Figure 1); however, some common patterns do emerge [20]. The results suggest that the induction of membrane deformation by a viral matrix proteins upon the recognition of charged phospholipids on membranes is a common feature of the assembly of enveloped viruses [23]. In addition, late assembly domains have been identified that link viral assembly to the cellular vesiculation machinery [24]. The major molecular pathway involved is facilitated by the endosomal sorting complexes required for transport (ESCRT) protein family. The acquisition of an envelope has been termed either ESCRT-dependent (HIV) or -independent (IAV) [24]. Upon the release of particles (or virions), morphological changes—due to proteolytic activity or environmental changes—become apparent as a consequence of virus maturation [25,26].

Additionally, other pathways play a role (e.g., via the Ras oncogene-related protein RAB31) [27]. However, dependency on these pathways may be varying and intermittent: different virus species may utilize a range of ESCRT-family proteins, while employing viral proteins (often matrix proteins located in between capsids and envelopes) for specific functions [24]. The complexity of various virus particles varies from only a few gene products (e.g., *Parvovoridae*, *Retroviridae*) to hundreds of gene products (e.g., *Herpesviridae*, *Poxviridae*). More complex virus species can afford more independence and not only bud independently of ESCRT but also as largely independent from any cellular proteins. Newcastle disease virus (NDV, *Paramyxoviridae*) contains a membrane-associated matrix protein which induces both the formation of the vesicle, but also its release (“scission”) [28]. Additionally, different subtypes of vesicles are usually produced. In poxviridae (Vaccinia virus/VACV), more than one envelope lipid bilayer may be present surrounding the core. Forms with a single membrane (mature virion) and double membrane (external virus) are released from the cell by lysis or exocytosis, respectively, and may serve different functions [29]. Hepadnaviruses produce a large number of non-infectious particles with smaller spherical and filamentous morphologies serving immune decoy functions. Newly formed viral particles may generally be categorized as infectious or non-infectious and thus constitute at least two different subtypes. 

For EXOs, biogenesis defines cargo and thus function. Consequently, a precise understanding of biogenesis is crucial for exploring the physiological complexity as well as exploiting the biotechnology potential of the exosome platforms [30,31,32,33]. Exosomes correspond to the intraluminal vesicles (ILVs) stored in multivesicular bodies (MVBs, also late endosomes). The endosomal pathway starts with the formation of early endosomes by the uptake of proteins, lipids, and other extracellular molecules at the plasma membrane [31,33]. This initiating step has been described as a micro-autophagy event [34]. As the endosomes enrich intraluminal vesicles in the lumen, they become MVBs or late endosomes. The most important pathway for MVB formation is the endosomal sorting complex required for transport (ESCRT) machinery. It consists of four complex proteins, namely ESCRT 0, I, II, and III, and the ATPase Vps4 that recognize and recruit cargo and enable membrane budding to form ILVs. ESCRT 0 initiates the formation of ILVs in early endosomes by interacting both with the endosomal membrane, namely the phosphatidylinositol 3-phosphate, and with the ubiquitinylated cargo proteins. ESCRT I and II complexes help with the clustering of the proteins and bind to ESCRT III, which attaches to the neck of the forming ILV and ATPase Vsp4 finally promotes the budding and detachment from the endosomal membrane [32,33]. Complex lipids such as ceramide also play an important role in ILV formation being part of the ESCRT independent pathway together with syntenin—ALIX, SNARE proteins, Rab GTPases, and tetraspanins CD9, CD63, and CD81, which are used as common exosomal markers due to their enrichment in exosomal membranes [31,32]. ESCRT-dependent and ESCRT-independent pathways seem to work synergistically [5]. MVBs communicate with other organelles such as the trans-Golgi network (TGN), the endoplasmic reticulum (ER), and the mitochondrion to modify the cargo [32]. MVBs are transported by microtubules or the cytoskeleton, either to the cell membrane to release the ILVs as exosomes to the extracellular space or they fuse with lysosomes and undergo degradation (see Figure 3). Rab GTPases regulate the intercellular transport, membrane budding, and fusion, and have a key role in the fate of MVBs [27]. To conclude, exosome biogenesis is a highly complex process that is dependent on cell type and environmental conditions.

Finally, the release of EXO marks a significant change in environment: from the acidic pH encountered in endosomes to the neutral pH state of the extracellular milieu. This event may constitute a type of maturation influencing the charge, structure, and function of EXO molecules facing the exterior, probably inducing an activating stimulus. The switch from pH5 to pH7 seems to increase membrane rigidity [35]. A link between the pH and levels of produced EXOs has been identified in the human cancer cells of different histotypes [36,37]. The increased acidity in different cancer cell types increases the levels of EXOs and this may apparently have a general effect on cancer cells [36].

For both EXOs and EVIs, specific membrane compartments (as well as lipid rafts (LRs), tetraspanin-enriched membranes (TEMs), and detergent-resistant membranes (DRMs)) [38,39,40] play significant roles in biogenesis. The concept suggests that specific areas of the membrane are given a certain function, as represented by a different composition (i.e., an enrichment in cholesterol and sphingolipids). The existence of these compartments has been highly controversial due to small spatial dimensions and short half-lives [39,40]. Essentially, they provide a platform for assembly as well as for the entry of viruses (e.g., SARS-Co-V-2). Proteins that often associate with these membrane compartments include glycosylphosphatidylinositol-anchored proteins, Flotilins, and different G-protein-coupled receptor proteins. Tetraspanins commonly found in VEN are also enriched in membrane microdomains and play an important role in virus entry and exit [38]. The biogenesis of EXOs and EVIs may influence each other. Viruses are able to usurp the EXO biogenesis pathways and thus influence the secretion of exosomes [24,41]. The expression of the Epstein–Barr virus (EBV) LMP1 protein induced an increase in EXO levels compared to that not expressing this protein [42]. Similarly, the human papillomavirus (HPV) E6/E7 protein affects the level or EXOs in HPV-positive tumors [43].

## 3. Viral Envelopes and Exosomal Membranes

While EVIs have evolved to have a certain degree of stability, EXOs are highly dynamic and variable. Several layers of diversity are encountered on the level of organism, tissue, cells and environmental conditions [44]. This diversity is reflected both in terms of cargo, membrane lipid, and protein composition. The dynamic nature may be a byproduct of biogenesis that reflects the cell state and may be communicated to neighboring cells. The producing cell sends a “status message” to the recipient cell. Both consist of a soluble (hydrophilic internal) fraction, delineated by a lipid (hydrophilic) bilayer. The functional priorities of the limiting membranes are similar for VENs and EXOs—the protection of content/cargo and the facilitation of cellular exit and entry. The similarities of EXOs and viral particles, especially concerning their surface characteristics (envelope vs. membrane), often allows for cooperation or competition. Viruses exploit exosomes to extend transmission range and subvert immunity [45] and a complex interaction pattern is beginning to evolve around the influence of the virus and exosome on tumor formation and progression [43,46]. A common hallmark of EXO and VEN membranes is the marked variation in the (membrane) composition compared to the source membranes with respect to both the vesicles’ lipidome and proteome.

In addition to the lipid composition or membrane protein content, other parameters can be used to characterize EVI and EXO membranes: curvature, stability (or rigidity), and the distribution of lipids between the two leaflets of a lipid bilayer (symmetry) (see Section 3.1) [3,47,48].

### 3.1. Lipid Content and Characteristics

During the biogenesis of the vesicles, lipid metabolism is tightly correlated [49]. Lipids serve as the structural building blocks of the membranes as well as play a significant role as an elements of the signaling pathways [50]. In the replication cycle of viruses, their influence can be found at various phases since they serve as (co-)receptors for entry. Replication and assembly are often linked to lipid bilayer structures and viral structural proteins are frequently post-translationally modified with lipid moieties [51]. Similar aspects may be playing a role at corresponding steps in the lifecycle (see Figure 2) in EXOs.

The finding that the lipid compositions of EXOs and VENs vary from the same source membrane suggests active exclusion and enrichment events during formation. In an early study by Aloia et al. from 1993, the membranes of uninfected cells were compared to the membranes of infected cells and viral particle membranes [52]. The results demonstrated that both the cell membranes and virus envelopes varied in lipid composition from uninfected cell membranes. Most notably, the content of phosphatidylcholine was significantly increased in the membrane of infected cells and reduced in the virions. This suggests a segregation of lipid species during vesicle formation. Additionally, the ratio of cholesterol to phospholipids (such as phosphatidylcholine) was increased [52]. Commonly enriched lipid species in EXOs are cholesterol, sphingomyelin and phosphatidylcholine [53,54]. For HIV VEN, a most significant increase in cholesterol compared to the source membrane is observed, alongside a high degree of order not typical of cellular membranes [52]. For Influenza, differences in lipid composition have been identified in different strains [55]. Environmental factors may also influence the envelope lipid composition. However, large comparative studies comparing the enveloped virus membranes of different species and their source cellular membranes are missing. Mechanisms governing such specifications may include physical constraints, issues of location, or active interaction processes (similar to protein incorporation processes). Compositional variations may help define subtypes [56] or facilitate diagnostic applications by referring to pathological conditions [57,58]. In the first case, three different variants of small EVEs (50–200 nm) were identified based on their affinity for specific lipid-binding compounds (annexin, shiga toxin B, cholera toxin B). In the latter, the influence of hyperglycemia on EVEs was examined. A change in the abundance of selected fatty acids could be demonstrated [57]. The purpose of the changed membrane composition is most likely a stability issue. Membranes may also be characterized by parameters other than composition such as stability. The organization of the EXO and EVI membranes (defined by composition and environmental factors) favors highly ordered phases of increased rigidity [52,59]. Essentially, this is a concession to the extracellular “lifestyle” and the protective functions of the membranes for EXO and EVI. Other parameters include the curvature of the vesicles and the symmetry between the inner and outer membrane leaflet. Naturally, the nanosized EXO and EVI require an increased curvature which is to be facilitated by protein–membrane interactions and lipid flip-flops [48,60,61,62]. The latter changes the relative composition of inner and outer membrane leaflets, affecting the membrane symmetry. Asymmetry between membrane leaflets refers to the fact that lipid species may be enriched in the inner or outer leaflet. While this switching occurs very slowly without active enzymatic support, cellular flippases, floppases, and scramblases mediate the lipid exchanges between membrane leaflets [48]. As a consequence, lipid species may be exposed to the exterior. This, in turn, can trigger further events, i.e., the activation of viral fusion proteins upon acidic flipping [62]. Exosomes lack phospholipid asymmetry [5,35], leading to a high content of phosphatidylserine (PS) and phosphatidylethanolamine (PE) on the vesicle surface, contributing to stability and additional attachment capabilities due to the exposed lipids [5,63].

Lipidomic technologies are emerging as powerful tools. Studies have, for example, been conducted to differentiate the lipid composition of EXOs and membrane vesicles from different cell types [44]. A clear difference in the lipid composition of the two subtypes of extracellular vesicles was observed. EXOs were enriched for glycolipids and free fatty acids, whereas membrane vesicles were enriched for ceramides and sphingomyelins [44]. Additionally, cell type-specific differences could be detected. In mesenchymal stem cells (MSCs), cardiolipins were enriched, and in the glioma cell line U87, sphingomyelins were enriched. Lipidomic studies have also been employed to differentiate pathological from physiological states. EXOs from ovarian cancer cells were compared to healthy tissue-derived EXOs [64]. Additionally, in such cases, differences could be observed, e.g., an increased amount of cholesterol ester and zymosterol were found in the cancer cells compared to normal tissue. Such studies are an important requisite for the use of EXOs as diagnostic tools. 

Summing up influences on lipid composition for both, we can identify three major aspects: source membrane composition, vesicle-specific influence (from vesicle protein–lipid interactions or the scrambling/flipping of lipids) and environmental factors (pH, metabolic state, pathology).

### 3.2. Protein Content

A range of proteins have been identified as being specifically enriched in exosomal membranes. Primary marker proteins are members of the tetraspanin family (CD9, CD63, CD81). Tetraspanins are regulators and facilitators of vesicle formation. As such, they also play a role in viral assembly [38]. The major difference between EXO and VEN proteomes is the (under naïve conditions) lack of viral proteins. However, in the case of infections, viral proteins may be quickly found in EXOs proving a strong relationship in terms of physical characteristics and the use of the same intracellular pathways [1]. Viral envelope proteins serve functions in attachment and entry. The major viral glycoproteins usually define the host range, i.e., which cellular receptors may be contacted and used to initiate entry. The expression distribution of these receptors will define whether these cells are “permissive” for viral infection. Similarly, the ensemble of membrane proteins will define the functional interactions that are possible for the EXO. In VENs, the coverage with viral proteins may be very dense, emulating a second capsid layer, e.g., retroviridae (HIV) display a more relaxed configuration than Flaviviridae, allowing the introduction of cellular proteins. These processes are at least partially regulated. While some proteins are enriched (HLA-DR, ICAM-1, PD-L1), others are decreased in level or even excluded (CD45, CD4, CD80) [65], The situation may be similar for EXO membranes. Mechanisms of incorporation include passive and active (through interactions between cellular and viral proteins) uptake and the co-localization at lipid rafts or membrane microdomain structures. Viral glycoproteins may be exchanged between different species (and families) (e.g., in Retroviridae), a mechanism or tool used by virologists known as pseudotyping. For viral envelopes, immune evasion constitutes another function. In its extracellular stage, the virus particle is exceptionally susceptible to interventions from the immune system, e.g., neutralizing antibodies, compliment). The removal of immunostimulatory elements from the envelope will actively contribute to this strategy. 

Proteomics analysis has been conducted on EXO preparations under various circumstances [44]. Common marker proteins such as the tetraspanins have been identified and an overview of cargo elements is emerging. More complex studies have been conducted, e.g., combining proteomics and lipidomic data for comparing EXOs and membrane vesicles. When comparing EXOs and membrane vesicle proteomes, EXOs were found to be enriched in extracellular matrix, heparin-binding-receptor, immune response, and cell adhesion functions. MVs were enriched for endoplasmatic reticulum, proteasome and mitochondrial proteins [44]. Interestingly, this demonstrates a bias towards an outward-bound or external mechanism for EXOs compared to more internal functions for MVs. Similarly, proteomics analysis has been conducted for the different species of enveloped viruses [66], also assessing the incorporation of cellular proteins [65]. Membrane-specific proteomics may be achieved by previous membrane preparations or the in silico analysis of comparative proteomics (or lipidomics) between VEN and EXO membranes.

Certain types of enveloped viruses (e.g., *Herpes*- or *poxviridae*) contain a protein-rich matrix in the space between the capsid and envelope. This tegument (e.g., in *Herpesviridae*, also lateral bodies in the case of *Poxviridae*) is released into the cell upon viral entry. The tegument contains proteins and small RNA species, again resembling the EXO content or cargo and providing an additional reservoir of elements re-directing cellular processes and responses.

## 4. Discussion—Implications for Applications

The field of exosome or extracellular vesicle research has recently made significant progress due, which has certainly been in part due to the interest in exploitation for medical and diagnostic products. Solving issues of definition, preparation, and analysis has consolidated the field to some degree. However, some concepts are still emerging. These include how such issues fit into overall inter-cell signaling, regulatory mechanisms, and limiting circumstances. Inherently, such aspects may only be researched in vivo or in complex cell culture or tissue models. The similarities to virus biology are astonishing [15,43,67,68], however, they may be more of a result of parallel development due to similar functional demands, rather than being based on the “hijacking” of exosome biogenesis. Additionally, it may very well be misleading to consider all enveloped viruses to be one entity. Different viral families (or even species) may exploit different pathways. This diversity in approaches is a general hallmark of the viral lifecycle and may well extend the relationship to exosome biology.

Viruses have a long history as biotech workhorses, mostly due to their ability to stably introduce genetic information into cells if required with high efficiency. As a consequence of the similarities and common features, exosomal preparations are also utilized to serve similar functions in delivery applications [69]. Viruses and exosomes are complex molecular machines delivering what is essentially a combination “package” of signals. Deciphering this package and translating it into a metabolic program is key for exosome signaling. For practical applications, sorting and prioritizing signals will be essential. In viral vectors, a major step is removing large parts of the viral genome and manipulating packaging signals Similarly, controlling a small RNA repertoire and modifying protein composition will improve the performance of exosomal vectors. An interesting question is whether a hierarchy of signals is observed in a physiological context, leading to priorities in functions or effects, i.e., by suppressing minor activities. A special role falls to the exterior face of the vesicle membranes since it is the first element to encounter potential interaction partners or targets. Modifications at this site can change the immunological behavior as well as targeting and may additionally provide effector functions. Such modifications may either be realized by the genetic engineering of the producing cells (by introducing peptide signals directing the recombinant proteins to MVB and ILVs) or, alternatively, modifications may be established on fully formed vesicles by membrane-specific modifications. Chemical modifications (click-chemistry-based, conjugation) are performed with adaptor systems (genetically engineering an anchor point for heterologous factors) or membrane-tropic elements such as function–spacer–lipid (FSL) constructs or glycosylphosphatidylinositol (GPI) anchoring [70]. The latter technology, termed molecular painting, has been used to modify viruses, mammalian exosomes, and bacterial outer membrane vesicles [70,71,72,73] and can provide labelling [71,73] and immune-protection [72] to modified vesicles. Delivery is not the only mode of action for EXOs and related secretome components. Applications are developed in many areas, including wound healing [74,75,76], cosmeceuticals [77], and tumor therapy [78].

In addition, this is similar to the identification of viral infections, whereby circulating virus particles are often collected by less invasive approaches than from the sites of pathogenic events (e.g., rabies is diagnosed by less invasive procedures, i.e., from sputum rather than brain biopsies). The recovery, detection, and analysis of exosomes from living organisms enable diagnostics strategies. “Liquid biopsies” constitute cellular material derived from organs or tissue in the form of extracellular vesicles that may be found circulating in blood or other body fluids [79]. Such methods can help prevent or decrease the number of invasive procedures, e.g., in tumor diagnostics, and help enabling larger-scale screening. Generally, techniques and strategies may be easily adapted between exosomes and enveloped virus particles.

## 5. Conclusions

Finally, we can extract critical parameters for assessing the similarities between EXO and EVI membranes:Physical: size, density, charge;Membrane structures: lipid composition and membrane proteins;Content and cargo: enrichment and exclusion;Lifecycle: cellular contributions and functions of membrane vs. initiation and perspectives;Functions: reprogramming of cell gene expression and metabolism.

Differences are observed in the biogenesis of EXOs and EVIs (see Figure 3), whereas the relationship of EXOs and EVIs to cells reveals similarities (see Figure 2). Similar needs and constraints most likely lead to shared physical and biochemical characteristics.

Research on viruses has been conducted for more than a century due to their biomedical and biotechnical potential as well as their pathogenic nature. Foremost, the diversity encountered in the viral lifecycle suggests that enveloped virus particles are more than just capsids acquiring a lipid shell via exosome biogenesis. Physical and functional analogies are significant. As a corollary, both applications and techniques (used in preparation and analysis) may be exchangeable: if this works with viruses, it may very well also work with exosomes and vice versa. Both virologists and exosome researchers may be well advised to consider knowledge from the other field both regarding basic biology and applied aspects.

## Figures and Tables

**Figure 1 membranes-13-00397-f001:**
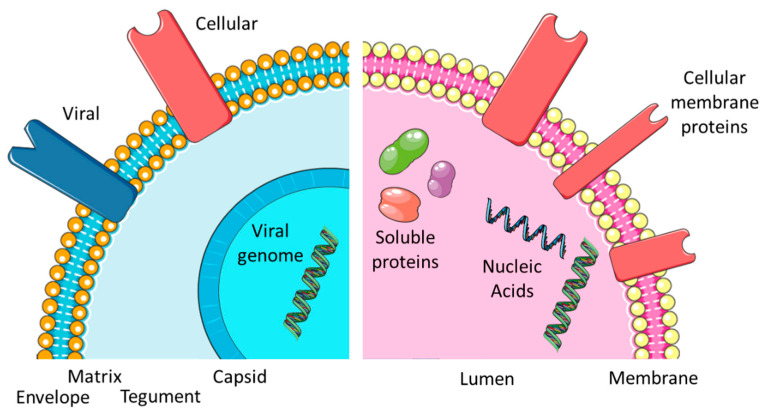
Comparing EXOs and EVIs. Overview of the virus and exosome morphology and content.

**Figure 2 membranes-13-00397-f002:**
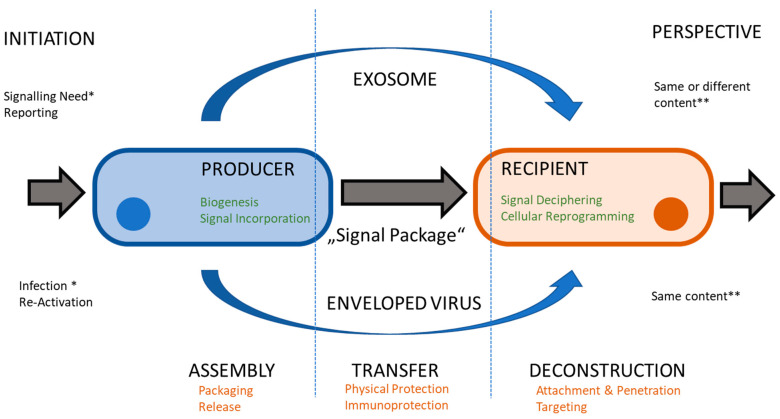
Comparing lifecycles. Both exosomes and virus particles transfer a package of signals from a producer cell to recipient cell. Cellular contribution (in green) and membrane functions (in orange) are similar, whereas initiating events (*) and perspectives (**) differ.

**Figure 3 membranes-13-00397-f003:**
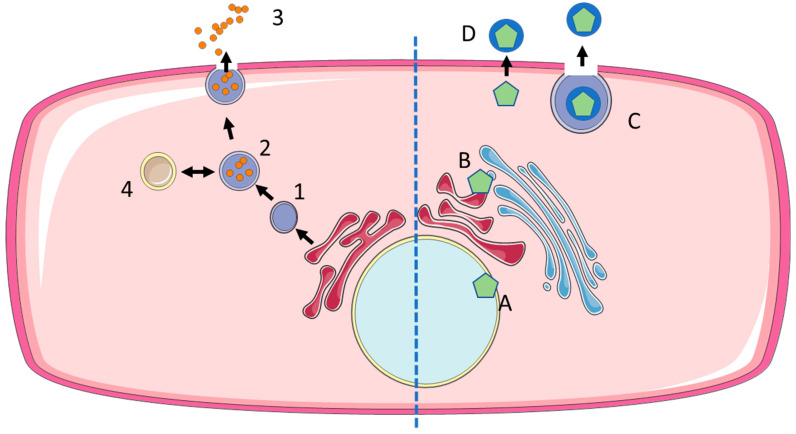
Biogenesis of exosomes and enveloped viruses. EXOs are produced from late endosomes (1) by the formation of interluminal vesicles (2), which are either released at the cell membrane to become exosomes (3) or recycled via fusion with lysosomes (4). Enveloped viruses acquire their envelope at different sites: nuclear envelope (A), endoplasmic reticulum or Golgi apparatus (B), followed by the release of exocytic vesicles containing virus particles (C). Alternatively, enveloped virus particles may acquire their envelope directly at the cell membrane (ectosomal budding) (D).

**Table 1 membranes-13-00397-t001:** a: Abbreviations and definitions of extracellular vesicles. b: Exosomal and viral components: definitions and abbreviations.

**Collective Term**	**Abbr.**	**Types**	**Abbr.**	**Subtypes**	**Variants**	**Relevance**
Extracellular vesicles ^a^	EVE	Ectosomes ^b^	ECT	Microvesicles (MIVs)		Signaling
				Apoptotic bodies (ABOs)		Apoptosis
		Exosomes ^c^	EXO	Based on size, cell type		Signaling
		Enveloped virus ^d^	EVI	Replication competent		Functional pathogen
				Replication incompetent	Naturally	e.g., immune decoy
					Artificial	e.g., gene therapy vectors
**Substructures**	**Abbr**	**Definition**				
Exosomal membrane	EXM	The protein-rich lipid bilayer surrounding an exosome				
Capsid	CAP	The highly organized protein lattice surrounding the viral genome				
Matrix	MAT	A protein-rich area found in some viruses connecting CAP and VEN				
Tegument	TEG	A protein and RNA containing structure found in some viruses, located between CAP and VEN				
Viral envelope	VEN	The protein-rich lipid bilayer surrounding a subset of virus species				

^a^ refers to the vesicular fraction of the cellular secretome. ^b^ refers to vesicles derived from the cell membrane. ^c^ refers to vesicles derived from internal membranes, more specifically the late endosome membranes giving rise to intraluminal vesicles (ILVs) within multivesicular bodies (MVBs). ^d^ refers to vesicles derived from either the cell or internal membranes, containing viral proteins and genome. Abbr. Abbreviation used in text.

**Table 2 membranes-13-00397-t002:** EXO and EVI characteristics.

GENERAL PROPERTIES AND MEMBRANE CHARACTERISTICS					
	Exosomes	HIV	IAV	VACV	SARS-CoV
**Diameter**	30–300 nm	80–100 nm	80–120 nm	220–450 nm long 140–260 nm wide	120 nm
**Marker proteins**	Tetraspanins (CD9, CD63, CD81)	gp120 (ENV), p24 (CA)	HA, NA,	NC	S(pike) or N(ucleocapsid)
**Cargo**	Cellular RNAs and proteins	Viral genome (ssRNA)	Viral genome	Viral genome	Viral genome
**Membrane origin**	Late endosome, multivesicular body	Plasma membrane	Plasma membrane	Endoplasmic reticulum, *trans*-Golgi	From Endoplasmic reticulum to Golgi-apparatus
**Biogenesis mechanism(s)**	ESCRT-dependent or independent	**MA**, (partially) ESCRT-dependent, alternatives	**MA2**, ESCRT-independent, alternatives	unusual, de novo lipogeneses	**M**, **N** and **E**,
**Difference to source membrane**	Increase in cholesterol, sphingomyelin, glycosphingolipids, phosphatidylserine. Decrease in phosphatiylcholine, phosphatidylinositol	Increase in cholesterol, decrease in phosphatidylcholine	Increase in cholesterol and sphingolipids; decrease in glycerophospholipids	Decrease in cholesterol; increase in phosphatidic acid and phosphatidylinositol	Decrease in cholesterol, increase in phospholipids
**Asymmetry**	unclear	Lost	Maintained	n.d.	n.d.
**Proteins, viral**	Facultative *	gp 120 or Env: SU and TM)	HA, NA, M2	Virion membrane proteins	S, M(embrane), E(nvelope)
**Proteins, cellular**	Tetraspanins (CD9, CD63, CD81)	excluded: CD45, CD4 enriched: HLA-DR, ICAM-1,	Yes	n.d.	n.d.

* Various viral proteins have been found in exosomes.

## Data Availability

Not applicable.

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
