# Peer review of "Extracellular Vesicles and Their Membranes: Exosomes vs. Virus-Related Particles"

_membranes, 2023, doi:10.3390/membranes13040397_

Round 1

Reviewer 1 Report

The current manuscript titled “Extracellular vesicles and their membranes: exosomes vs. virus related particles” by Galvez et. al., describes how various viruses assemble either at the membrane or in some cases in the nucleus.  They also describe the variety of the lipids in the EVs or viruses including commonly enriched lipid species in EXOs including cholesterol, sphingomyelin and phosphatidylcholine. The paper also describes proteins that are often associated with these membrane compartments include glycosylphosphatidylinositol anchored proteins, Flotilins and different G-protein-coupled receptor proteins. Overall, I found the paper interesting and novel in putting together relevant manuscripts for a review, however, there were some concerns, including:

1.       The manuscript needs to be read and edited!  There are places where the some of the references are not cited, i.e., “ESCRT-dependent (HIV) or -independent (IAV) (ref)……for specific functions (Ref)”.

2.       It was not clear if viral membranes also have tetraspanins (how many and potential ratios) and if the EV membranes are flipped (inside-out) prior to exit from cells?

3.       The changes in pH before and after exit for EVs needs to be described better.  Molecular mechanisms that control the pH alterations needs to be developed.  

Author Response

We would like to thank the reviewer for the detailed review and constructive comments. Please find the replies to reviewer’s comments below (in italics)

(x) English language and style are fine/minor spell check required

The current manuscript titled “Extracellular vesicles and their membranes: exosomes vs. virus related particles” by Galvez et. al., describes how various viruses assemble either at the membrane or in some cases in the nucleus.  They also describe the variety of the lipids in the EVs or viruses including commonly enriched lipid species in EXOs including cholesterol, sphingomyelin and phosphatidylcholine. The paper also describes proteins that are often associated with these membrane compartments include glycosylphosphatidylinositol anchored proteins, Flotilins and different G-protein-coupled receptor proteins. Overall, I found the paper interesting and novel in putting together relevant manuscripts for a review, however, there were some concerns, including:

  1. The manuscript needs to be read and edited! There are places where the some of the references are not cited, i.e., “ESCRT-dependent (HIV) or -independent (IAV) (ref)……for specific functions (Ref)”.

We have edited and corrected the manuscript. References are now fully in place (up from 50 to 80, see reference list). The text has been checked by external review.

  1. It was not clear if viral membranes also have tetraspanins (how many and potential ratios) and if the EV membranes are flipped (inside-out) prior to exit from cells?

We have addressed the issue in sections (Biogenesis, membrane). In brief tetraspanins are an important aspect also of viral biogenesisand membrane flipping events influencing membrane symmetry do occur in EVs and Exos

  1. The changes in pH before and after exit for EVs needs to be described better. Molecular mechanisms that control the pH alterations needs to be developed. 

We have tried to include additional information. Microenvironmental pH changes play a significant role in physiological and pathological context.

Reviewer 2 Report

Daniela Cortes-Galvez and colleagues reviewed similarities and differences between exosomes and enveloped viral particles focusing on the membranes features. The matter is foremost important and news are always of great interest. Although the good attempt of the authors to make a comparison between viral membrane particle and exosome, the argument need additional information to improve its relevance. Thus the following points have to be approached.

Main points

1- Introduction: the authors focused on the exosomes but a little explanation of the other extracellular particles could be of relevance. The role of other extracellular vesicles is discussed for many viruses in literature as well as exosomes thus an overview of these alternative vesicles are important.  In this regard, at line 27 the authors have to explain the subject (Exosomes?) because is not clear.

2- Section 2. The provenience of viral envelope membranes has to be better described to explain the difference (if there are any) in the molecular composition of the membranes from different provenance.

3- Section 3. The point 3.2 Lipid Content (it should be changes in point 3.1) describe the lipid species present in exosomes. For the relevance of this point in the review argument, a table reporting such information in exosomes, other extracellular vesicles and different viral membranes could be important. Moreover, the different resistance of exosomes membrane compared to the other viral enveloped particles could be relevant to dissect the choice of virus to use the extracellular vesicles instead of enveloped. Can the authors add this information?

4-Discussion. The authors should report the relevance for the viruses to use extracellular vesicles membrane in their life cycle, replication and transmission, immune escape to persist in the host.

Minor points

1- lines 30, 35, 112, 115…. What is the meaning of ref? Additional reference?

2- Figure 1. All the abbreviation should be reported with full name in the foot notes of table/figure.

3- Line 72. What is EVI in this context? Extracellular vesicles membranes?

4- Figure 3. In the legend it should be correct point 6 (&).

5. Section 3.1 should be change in 3.2 or move to the correct point.  

Author Response

We would like to thank the reviewer for the detailed review and constructive comments. Please find the replies to reviewer’s comments below (in italics).

(x) I don't feel qualified to judge about the English language and style   

Comments and Suggestions for Authors

Daniela Cortes-Galvez and colleagues reviewed similarities and differences between exosomes and enveloped viral particles focusing on the membranes features. The matter is foremost important and news are always of great interest. Although the good attempt of the authors to make a comparison between viral membrane particle and exosome, the argument need additional information to improve its relevance. Thus the following points have to be approached.

Main points

1- Introduction: the authors focused on the exosomes but a little explanation of the other extracellular particles could be of relevance. The role of other extracellular vesicles is discussed for many viruses in literature as well as exosomes thus an overview of these alternative vesicles are important.  In this regard, at line 27 the authors have to explain the subject (Exosomes?) because is not clear.

We have extended the introductory parts, spending more place on overall EV descriptions and extending the range including VLPs. This has come about because some of the older literature suffers from unclear definitions of terms that have later been consolidated in the field. Table 1 attempts to clarify this issue.

2- Section 2. The provenience of viral envelope membranes has to be better described to explain the difference (if there are any) in the molecular composition of the membranes from different provenance.

We have tried to include additional information on the topic. Viral lipid envelope composition seems to be determined by source membrane, membrane compartments (DRMs, TEMs), virus-specific processes (e.g. through interactions with viral proteins) and environmental factors. Dominant elements (i.e., cholesterol) may be identified but comparison is difficult and would require direct comparative studies. What emerges is a set of membrane properties: increased curvature, increased stability (rigidity) and changed leaflet symmetry.

3- Section 3. The point 3.2 Lipid Content (it should be changes in point 3.1) describe the lipid species present in exosomes. For the relevance of this point in the review argument, a table reporting such information in exosomes, other extracellular vesicles and different viral membranes could be important. Moreover, the different resistance of exosomes membrane compared to the other viral enveloped particles could be relevant to dissect the choice of virus to use the extracellular vesicles instead of enveloped. Can the authors add this information?

We have tried to include additional information on lipid composition and the factors that can influence this composition. In addition, we have added information on the redistribution of lipids in the bilayer leaflets.

4-Discussion. The authors should report the relevance for the viruses to use extracellular vesicles membrane in their life cycle, replication and transmission, immune escape to persist in the host.

We have tried to include additional information. However, a full treatise of the issue would exceed the scope of the review. References to several reviews on the topic are included.

Minor points

1- lines 30, 35, 112, 115…. What is the meaning of ref? Additional reference?

We have added the missing additional references.

2- Figure 1. All the abbreviation should be reported with full name in the foot notes of table/figure.

We have summarized abbreviations and term descriptions in table 1.

3- Line 72. What is EVI in this context? Extracellular vesicles membranes?

Enveloped Virus. We have summarized abbreviations and term descriptions in table 1.

4- Figure 3. In the legend it should be correct point 6 (&).

We have corrected the error.

  1. Section 3.1 should be change in 3.2 or move to the correct point.

We have corrected the error.

Reviewer 3 Report

The manuscript attempts to summarize information about extracellular vesicles and virus-like particles. The major issues of the manuscript include:

(1) The authors for some reason compare viral particles and extracellular vesicles. VLPs are not viruses, there is no reason to provide information about virions of different viruses, and compare them with nanovesicles of cellular origin. The sole type of VLPs that should be given here and be compared with EVs is the VLPs based in gene editing and transfer of biomolecules for therapeutic purposes. The rest of the information is not relevant. Viruses and EVs should not be comapred in the first place.

(2) The term "extracellular vesicles" is not properly provided and misleading throughout the manuscript. 

(3) many references are missing. "ref" signs are given at some places, which apparently point to the places where references should be added.

(4) English language is poor and requires extensive editing.

(5) Figures are poorly drawn, not informative and not helpful to the readers.

Author Response

(x) Extensive editing of English language and style required        

Comments and Suggestions for Authors

The manuscript attempts to summarize information about extracellular vesicles and virus-like particles. The major issues of the manuscript include:

(1) The authors for some reason compare viral particles and extracellular vesicles. VLPs are not viruses, there is no reason to provide information about virions of different viruses and compare them with nanovesicles of cellular origin. The sole type of VLPs that should be given here and be compared with EVs is the VLPs based in gene editing and transfer of biomolecules for therapeutic purposes. The rest of the information is not relevant. Viruses and EVs should not be comapred in the first place.

The authors respectfully disagree with the referee’s opinion. Alone the number of publications and reviews available on the topic, many of which we have referenced here, clearly show we are not alone with this understanding. We believe there is a strong relevance in the comparison of cell derived lipid bilayer-enclosed vesicles of sub micrometer size containing protein, RNA and/or DNA, transferring information from a producer to a recipient cell, with the explicit aim of causing a change in recipient cell behavior. This definition, in our opinion, covers exosomes, viral particles, as well as virus like particles of natural or artificial nature. While we do not support any hypothesis on common ancestry of virus and exosomes, we observe a large number of correlations and similarities. These include applications, preparation and analysis technology, biogenesis overlaps, functional synergy and antagonism, pleiotropy, contribution to the (same) pathologies (i.e. tumors) as well as biochemical and biophysical similarities. We also acknowledge fundamental differences, i.e. evolutionary aspects (the need of viral particles to maintain a degree of stability in order to protect the genetic information within vs. the high degree of variation in exosomes).

(2) The term "extracellular vesicles" is not properly provided and misleading throughout the manuscript.

We have included and overview of used abbreviations and term definitions (see table1). We have now defined extracellular vesicles as the vesicular fraction of the secretome.

(3) many references are missing. "ref" signs are given at some places, which apparently point to the places where references should be added.

We have added the missing references.

(4) English language is poor and requires extensive editing.

We have rechecked the text for further improvement and recruited external editing.

(5) Figures are poorly drawn, not informative and not helpful to the readers.

We have re-worked the figures and hope they provide significant added value. The prime aim of the figures is to allow a direct comparison between virus and exosomes.

Round 2

Reviewer 1 Report

The authors have responded well to my comments.

Author Response

We would like to thank the reviewer for the time and consideration. We have subjected the manuscript to another round of editing, identifying few issues (see manuscript in correction mode). We believe that the manuscript is now fit for publication.

Reviewer 2 Report

Authors have substantially improved the manuscript. Few typos remain to be corrected (es. Table 2 N(ucleocapsid) under SARS-CoV-2).  Now the manuscript is suitable for publication.

Author Response

We would like to thank the reviewer for the time and considerations. We have corrected the typo in Table 2 and subjected the manuscript to another round of editing, identifying few issues (see manuscript in correction mode). We believe that the manuscript is now fit for publication.

Reviewer 3 Report

Despite the technical improvements in the manuscript, I'm not convinced in the rationale of comparing EVs with viruses as well as disagree with the obscure similarities between them raised by the authors.

Author Response

We would like to thank the reviewer for the time and consideration. We have subjected the manuscript to another round of editing, identifying few issues (see manuscript in correction mode). We believe that the manuscript is now fit for publication. As to the unsolved issue, whether comparison of virus and exosome particles is applicable and acceptable, we would like to continue the discussion, but probably most useful using another platform. Please feel free to contact me via corresponding author's information.